

# Confidence in the dynamic spread of epidemics under biased sampling conditions

James Brunner and Nicholas Chia

Center for Individualized Medicine, Department of Surgery, Mayo Clinic, Rochester, MN, USA

## ABSTRACT

The interpretation of sampling data plays a crucial role in policy response to the spread of a disease during an epidemic, such as the COVID-19 epidemic of 2020. However, this is a non-trivial endeavor due to the complexity of real world conditions and limits to the availability of diagnostic tests, which necessitate a bias in testing favoring symptomatic individuals. A thorough understanding of sampling confidence and bias is necessary in order make accurate conclusions. In this manuscript, we provide a stochastic model of sampling for assessing confidence in disease metrics such as trend detection, peak detection and disease spread estimation. Our model simulates testing for a disease in an epidemic with known dynamics, allowing us to use Monte-Carlo sampling to assess metric confidence. This model can provide realistic simulated data which can be used in the design and calibration of data analysis and prediction methods. As an example, we use this method to show that trends in the disease may be identified using under 10,000 biased samples each day, and an estimate of disease spread can be made with additional 1,000–2,000 unbiased samples each day. We also demonstrate that the model can be used to assess more advanced metrics by finding the precision and recall of a strategy for finding peaks in the dynamics.

## INTRODUCTION

Policy decisions in the face of an epidemic rely on perceptions of the population dynamics of an infectious disease, for example, whether cases are growing or shrinking (*Centers for Disease Control & Prevention, 2020*; *Occupational Safety & Health Administration, 2020*; *Lee et al., 2020*). If everyone were tested every day, then this would be a matter of looking at the trend in the total number of positive tests per day. However, this scenario is unrealistic. In actuality, the population sampling needed to track an epidemic in a community will depend on the nature of the question we would like to answer. Sometimes, these questions are in conflict with each other. For instance, the primary goal of healthcare providers is to identify infected patients in the hospital or clinical setting so that appropriate treatment and protective measures may be prescribed, while at the other extreme the epidemiologist concerned with infection prevention within the population may be interested in determining the number of infected individuals in the population in

Corresponding author
James Brunner,
brunner.james@mayo.edu

order to focus efforts at limiting disease spread. Clearly, the former is very targeted towards patients showing up at a clinic with certain symptoms while the latter requires broad testing of both symptomatic and asymptomatic populations.

The reality is that testing needs to serve multiple purposes with a finite number of tests. In particular, the COVID-19 pandemic and response from world leaders has shed light on the need for a better understanding of community infection data and how to use it, both for decision making and media reporting. Some of the particular challenges are the significant proportion of asymptomatic carriers of the disease (*Bai et al., 2020*; *Mizumoto et al., 2020*; *Pollán et al., 2020*) and the changing availability of the testing. Both have resulted in testing that is strongly biased towards infected individuals and not representative of the proportion of cases in the population. In this manuscript, we focus on two intermediate use cases—individuals and businesses that need to estimate risk, that is, the probability that an infected person will be present in a given situation, and public policy makers that need to understand changing trends in the spread of the disease. We show that this can be done with a combination of biased and unbiased sampling that requires many fewer tests to be performed every day, but importantly must include the number of negative tests in addition to the number of positive cases that is more widely reported. Notably, this is in line with the World Health Organization's global surveillance guidelines, which include reporting of total tests so positive percentage can be determined (*World Health Organization, 2020*).

The purpose of this manuscript is to introduce a method for assessing confidence in conclusions made from biased sampling of the spread of an epidemic, and therefore providing a tool in calibration of data analysis and prediction methods. We begin with a calculation of the number of tests needed to identify a significant portion of infected individuals in a given day. We then describe a stochastic dynamical model that simulates testing over the course of an epidemic with known dynamics. We then show how we can use a given model of epidemic dynamics to investigate the amount of testing needed to estimate disease spread and trends in disease. We further use our approach to simulate testing with variable bias and error and investigate the roles of bias and error in testing. We find that amount of testing needed to identify most infected people in a population of 300 million (approximately the population of the USA), is extremely high. On the other hand, we show that trends in the spread of the disease can be accurately identified by sign (i.e., positive or negative) with less than 10,000 biased tests per day. We show that approximately 1,000 additional unbiased tests can be used to estimate the bias in testing. This can be used to estimate the extent of disease spread in the community on a daily basis. Our approach can also be used to assess the reliability of many data analysis techniques. We demonstrate this by assessing a strategy for finding peaks in the dynamics of the outbreak by using a smoothed numerical derivative of the data. Finally, we show the importance of understanding bias under the conditions of limited testing by examining COVID-19 data from within the USA.

As a special note, we emphasize that while many important efforts are being made to model the spread of COVID-19 and determine how testing can be used to reduce that spread (*Reich et al., 2020*; *Piguillem & Shi, 2020*; *Alvarez, Argente & Lippi, 2020*;

*Chowdhury et al., 2020*), our approach does not attempt to predict disease spread. Instead, we are testing confidence in data analysis sampled from known dynamics. In other words, rather than trying to predict the future, our work focuses on estimating confidence in current trends under non-ideal conditions.

## METHODS

### Sampling all infected

Sampling the population in order to identify all or some large proportion of the infected individuals will require a large amount of testing. If we assume that testing is done in a single day, the proportion of infected in the population is roughly constant in that day, and no person is tested more than once, the number of positive cases will follow a hypergeometric distribution based on the number of cases in the population, the bias in the testing and the number of tests performed. We can compute the cumulative distribution function, and so compute the number of tests needed so that the probability that some number of cases are found.

We take testing to be a process of sampling with replacement in a population of size $T$. To do this, we must compute the number of infected people in the population, which is given by $rT$. In principle, we may use a hypergeometric distribution with population $T = rT + (1 - r)T$ and infected $rT$. However, this approach assumes no bias in the testing, meaning that the probability of each individual (symptomatic or not) being tested is uniform. In the real world, symptomatic individuals are much more likely to be tested for a disease. We therefore introduce non-dimensional a bias parameter $B$ and take the apparent number of infected to be $rBT$, so that the apparent total population is $rBT + (1 - r)T$. For $B > 1$, this biases the testing towards infected individuals, representing the fact the symptomatic individuals are more likely to be tested, and also more likely to be infected. The rest of this paper considers evaluating the infected population under biased testing conditions as a means to understand trends in real data.

In our simulations of biased testing, we use a bias on the order of $B = 10$. However, this is done only to illustrate the method and should not be taken as an estimate of the bias from real data. We discuss below how bias may be estimated from a combination of biased and unbiased samples.

### A stochastic model for disease sampling

We developed a stochastic model to simulate the sampling of a population that is undergoing an epidemic with known dynamics. That is, given an underlying set of dynamics tracking asymptomatic infected individuals, symptomatic infected individuals, and non-infected individuals, we simulate testing members of this population for the disease. Let $I^1(t)$, $I^2(t)$ and $H(t)$ represent the number of asymptomatic infected individuals, symptomatic infected individuals, and non-infected individuals in the population, respectively, for $t \in [0,T]$. Our model does assume that the dynamics of the disease spread are some known functions of three compartments. In practice, such

dynamics are often simulated by simple systems of ordinary differential equations. However, this need not be the case, and dynamics can even be given directly as functions of time.

Tests are assumed to be carried out according to a Poisson point process (see *Klenke (2014)* and *Anderson & Kurtz (2011)* for a detailed introduction) with (possibly time varying) intensity function $\lambda(t)$. The intensity function $\lambda(t)$ can be interpreted as the rate at which members of the population are tested for the diseased which is assumed to be known. For example, an increase in test availability would be reflected in this model with an increase in $\lambda(t)$. Additionally, the incidence of positive tests being carried out is itself a Poisson point process with intensity $\lambda^+(t)$, and likewise the incidence of negative tests is a Poisson point process with intensity $\lambda^-(t)$, subject to the relation $\lambda(t) = \lambda^+(t) + \lambda^-(t)$. In this manuscript, $\lambda(t)$ is taken to be constant unless otherwise noted.

Each time a test is performed, we determine the status of the person tested. Under the assumption of unbiased testing, we categorize a person into one of three pools. The probability of a test result depends on the respective proportion of the population which belongs to each pool. That is, the probability the unbiased test is performed on an asymptomatic infected person is

$$P(\text{this testee is asymptomatic infected}) = \frac{I^1(t)}{I^1(t) + I^2(t) + H(t)} \tag{1}$$

Likewise, the probability that an unbiased test is performed on a symptomatic infected person is

$$P(\text{this testee is symptomatic infected}) = \frac{I^2(t)}{I^1(t) + I^2(t) + H(t)} \tag{2}$$

and the probability that an unbiased test is performed on a non-infected person is

$$P(\text{this testee is non-infected}) = \frac{H(t)}{I^1(t) + I^2(t) + H(t)} \tag{3}$$

We also account for the possibility that a test may give a false-positive or false-negative with some constant probability. Let $\varepsilon_1 \in [0,1]$ be the false-negative probability of the test and $\varepsilon_2 \in [0,1]$ be the false-postive probability. Then, for any given test we can combine Eqs. (1)–(3) to see the following:

$$P(\text{positive test}) = (1 - \varepsilon_1)P(\text{person is infected}) + \varepsilon_2 P(\text{person is not infected}) \tag{4}$$

$$P(\text{negative test}) = \varepsilon_1 P(\text{person is infected}) + (1 - \varepsilon_2)P(\text{person is not infected}) \tag{5}$$

Testing for disease in a population is not done uniformly at random. Instead, an individual displaying symptoms of the disease is much more likely to be tested for it than one who is not. We may model a bias in testing by adjusting Eqs. (1)–(3). Let $B(t)$ be some function of time, with $B(t) \geq 1$ for all $t \in [0,T]$. Then, we can reflect the bias of the testing

procedure by re-weighting the population for each test performed. We replace Eqs. (1)–(3) with the following:

$$P(\text{this testee is asymptomatic infected}) = \frac{I^1(t)}{I^1(t) + B(t)I^2(t) + H(t)} \tag{6}$$

$$P(\text{this testee is symptomatic infected}) = \frac{B(t)I^2(t)}{I^1(t) + B(t)I^2(t) + H(t)} \tag{7}$$

$$P(\text{this testee is non-infected}) = \frac{H(t)}{I^1(t) + B(t)I^2(t) + H(t)} \tag{8}$$

and combine Eqs. (6)–(8) with Eqs. (4) and (5) to determine the probability of a single test result. The result is that the number of positive and negative tests that have been carried out up to time $t$ are each non-homogeneous Poisson point processes with intensity functions

$$\lambda^+(t) = \lambda(t)\left[(1 - \varepsilon_1)\left(\frac{I^1(t)}{I^1(t) + B(t)I^2(t) + H(t)} + \frac{B(t)I^2(t)}{I^1(t) + I^2(t) + H(t)}\right)\right.$$
$$\left. + \varepsilon_2\left(\frac{H(t)}{I^1(t) + B(t)I^2(t) + H(t)}\right)\right] \tag{9}$$

$$\lambda^-(t) = \lambda(t)\left[\varepsilon_1\left(\frac{I^1(t)}{I^1(t) + B(t)I^2(t) + H(t)} + \frac{B(t)I^2(t)}{I^1(t) + B(t)I^2(t) + H(t)}\right)\right.$$
$$\left. + (1 - \varepsilon_2)\left(\frac{H(t)}{I^1(t) + B(t)I^2(t) + H(t)}\right)\right]. \tag{10}$$

We note that the model described above essentially assumes an infinite total population size. Practically, this means that testing is done on an insignificant proportion of the population, or equivalently that members of a population are immediately eligible to be re-tested after being tested. In Appendix E, we describe a modification for this model which accounts for small population size and non-immediate retesting.

### Simulation with the stochastic simulation algorithm

Both the initial model described in this manuscript and the model adjusted for small populations can be written as the sum of Poisson point processes with time-varying intensities. They can therefore be simulated using a slight adjustment to the Stochastic Simulation Algorithm (*Gillespie, 1976*, *1977*). This adjustment accounts for possible changes in the intensity functions between points in the Poisson processes, for example from variations in $\lambda(t)$ or $B(t)$. This adjustment is made by choosing event times according to the maximum values of any time-varying functions, and allowing for the possibility of a non-event at each event time, a procedure often called thinning (*Asmussen & Glynn, 2007*; *Anderson & Yuan, 2019*).

Simulation of the models as described allow us to perform Monte-Carlo estimations of the confidence that can be assumed in the calculation of various statistics from data. We perform Monte-Carlo estimation by repeatedly simulating sampling over the course of an epidemic with given dynamics and determining the success rate or average error of in determining a metric from simulated sampling when compared to determining the same metric from the known underlying dynamics.

## Estimating trends in disease spread

To assess trends in data simulated according to our model, we discretize the time interval $[0,T]$ into evenly spaced intervals (e.g., into single day increments) ending at times $t_1$, $t_2$, …, $t_N = T$. We then compute positive-test proportions for these intervals, simply defined as the proportion of tests carried out within the interval that were positive. This allows us to make sense of the simulated data even as testing capacity $\lambda(t)$ varies in time.

We estimate $N$-day trends in disease spread using a linear least-squares fit to $N$ consecutive days of simulated positive test proportions. We define the linear trend of the simulated data to be the slope of this fitted line. This can then be compared to a linear fit to the infected proportion

$$\frac{I^1(t) + I^2(t)}{I^1(t) + I^2(t) + H(t)}$$

over the same time interval, computed using time-discretized dynamics.

## Finding peaks in disease spread

We attempt to find peaks in the data by estimating the time derivative of the positive-test proportions simulated. We then identify peaks as points at which the derivative crosses from positive to negative. That is, we estimate the change in true positive-test proportion from day to day, and identify when this proportion stopped increasing and began to decrease.

To estimate the time derivative of positive proportions, we first compute a numerical derivative over each time interval. We then blur this discretized derivative to reduce noise (and therefore false peaks) using a one-dimensional Gaussian filter.

## Underlying dynamics

Our model is designed to simulate sampling of an epidemic with any non-negative underlying compartmental dynamics. This means that the number of healthy, symptomatic infected, and asymptomatic infected members of a population can be any non-negative known functions of time. As a consequence of this feature, some set of known dynamics must be either chosen directly or generated by another dynamic model. Confidence in metric sampling is then measured against the known dynamics.

To demonstrate our method, we use the SIR model (*Edelstein-Keshet, 2005*; *Hethcote, 2000*; *Kermack & McKendrick, 1927*) with a time-variable rate of disease spread to generate underlying dynamics, as well as a similar compartmental model that allows for asymptomatic individuals, which we refer to as the SAIR model. See Appendix D for

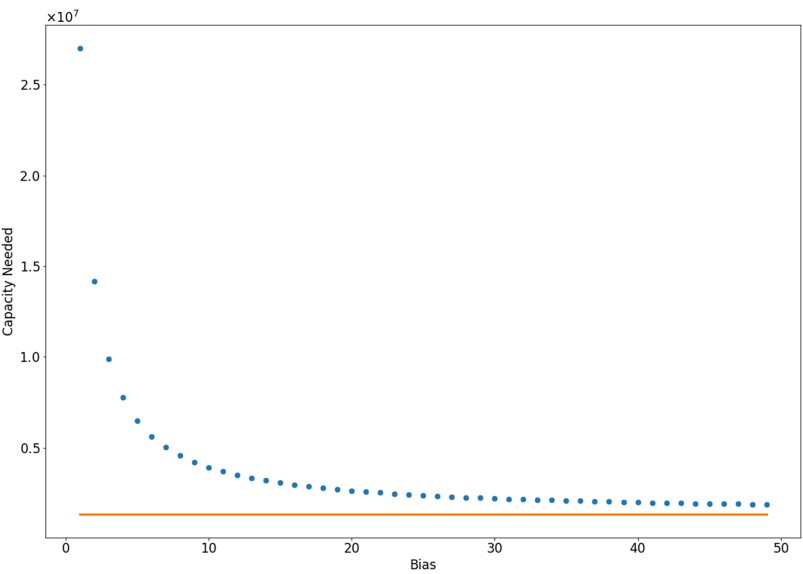

**Figure 1 Number of samples needed to identify 90% of infected individuals with 90% confidence, computed using a hypergeometric distribution.** In orange, we show the limiting case in which every person tested is infected, which can be interpreted as infinite bias. In this case, 1,350,000 tests are necessary.

details. These models represent a popular, simple choice of dynamic epidemic model with parameters often reported by the lay news media.

## RESULTS

### Sampling all infected

One obviously crucial role of testing of a disease outbreak is to identify patients for treatment and possible quarantine. From that perspective, it is crucial to identify all or most infected members of a community. We therefore assess the number of tests this would require, noting that these tests must be performed in a small enough time frame so that epidemic dynamics and bias in testing can be assumed constant. Using the cumulative distribution function of the hypergeometric distribution, we see that in a community of 300 million with 5% infection, we need approximately 27,009,300 unbiased tests in a short time interval to have 90% confidence that we have found 90% of the cases. For higher bias, fewer tests are needed, as seen in Fig. 1. This is likely an intractable number of tests to be performed in a short enough time interval (perhaps 1 day) so that epidemic dynamics and bias in testing can be assumed constant. In fact, during the COVID-19 epidemic of 2020, about 500,000–600,000 tests were performed each day across the United States by the end of June (*Lipton, Ellington & Riley, 2020*).

### Simulated sampling

Rather than attempt to find all COVID-19 patients, we may wish to simply have an accurate estimate of the number of infected. In order to asses our ability to do this, we simulated sampling as described above with underlying dynamics generated by ODE models of outbreaks. We use the positive test proportion per day as our sampled variable,

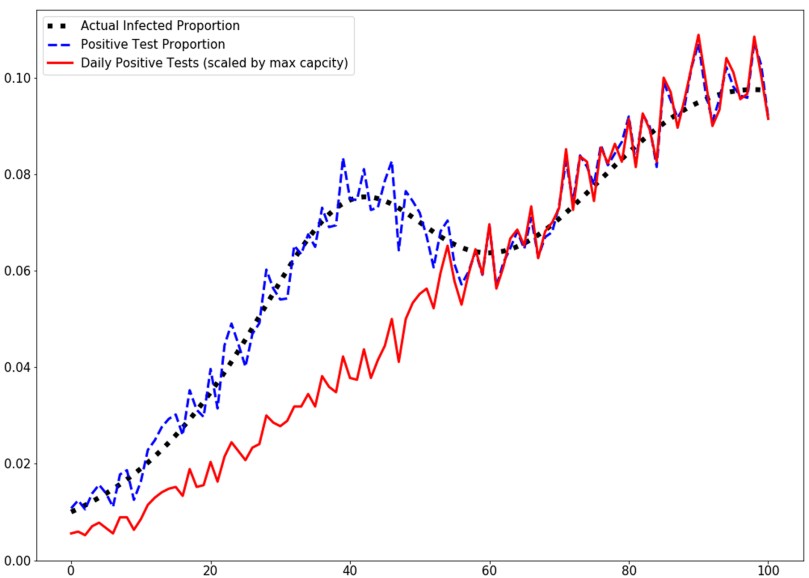

**Figure 2 Positive proportion of tests and scaled positive tests per day.** Testing capacity is initially 1,350 tests per day but rises to 2,700 midway through the simulation.

in order to account for variations in testing capacity. Figure 2 demonstrates that day–day variations in testing capacity can mask the real dynamics when only positive test count is considered. In contrast, the positive proportion of tests does captures the dynamics. In this simulation, we used a time-varying intensity $\lambda(t)$ which was taken to be

$$\lambda(t) = 0.75 + 0.25 \tanh\left(\frac{t - 50}{5}\right) \tag{11}$$

## Biases in testing

### Biased sampling

With false positive and false negative rates equal to 0, biased sampling causes an overestimation in the proportion of a large population that is infected. If tests are taken at random in the population (and so not biased), the proportion of positive tests will on average be the same as the proportion of infected individuals $r(t)$, which is the sum of the proportions of symptomatic and asymptomatic infected individuals:

$$r(t) = \frac{I^1(t) + I^2(t)}{I^1(t) + I^2(t) + H(t)} = \frac{I^1(t)}{I^1(t) + I^2(t) + H(t)} + \frac{I^2(t)}{I^1(t) + I^2(t) + H(t)}$$
$$= r_{I^1}(t) + r_{I^2}(t). \tag{12}$$

Biasing the tests towards symptomatic individuals is analogous to sampling a population with extra symptomatic individuals added:

$$r_B(t) = \frac{Br_{I^1} + r_{I^2}}{1 + (B - 1)r_{I^1}} > r(t) \tag{13}$$

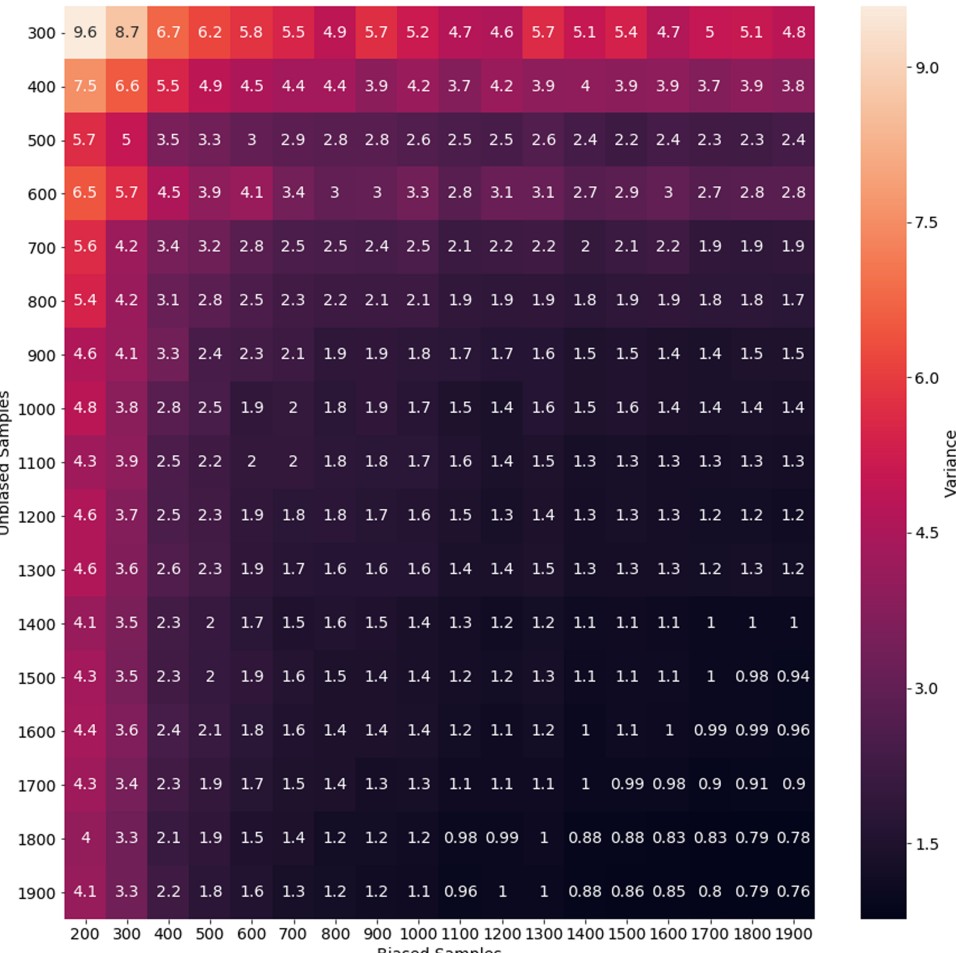

**Figure 3 Variance of bias estimate for various sampling rates.** This variance represents the error in estimation from a single day of biased and unbiased tests. In this experiment, we do not have asymptomatic infected individuals.

and we note that this overestimation will not lessen with a higher number of samples. Including the rate of false positive and negative tests, this becomes

$$r_B(t) = \frac{Br_{I^1}(1 - \varepsilon_1) + r_{I^2}(1 - \varepsilon_1) + \varepsilon_2(1 - r_{I^1} - r_{I^2})}{1 + (B - 1)r_{I^1}} > r(t) \tag{14}$$

In order to estimate the spread of the disease in a community (i.e., estimate the percentage of the population which is infected), we can estimate the bias $B$ if we have unbiased sample data as well. To do this with only positive/negative test data, we must assume that the ratio of symptomatic to asymptomatic infected members of the community is constant (i.e., $r_1 = cr_2$). Here, we are making an estimation analogous to a Monte-Carlo method, and so for better accuracy we need to reduce the variance in our estimate of $B$. We simulated an estimate of $B$ with various biased and unbiased sample capacities. Variances for these estimates are shown in Fig. 3. We see there that with

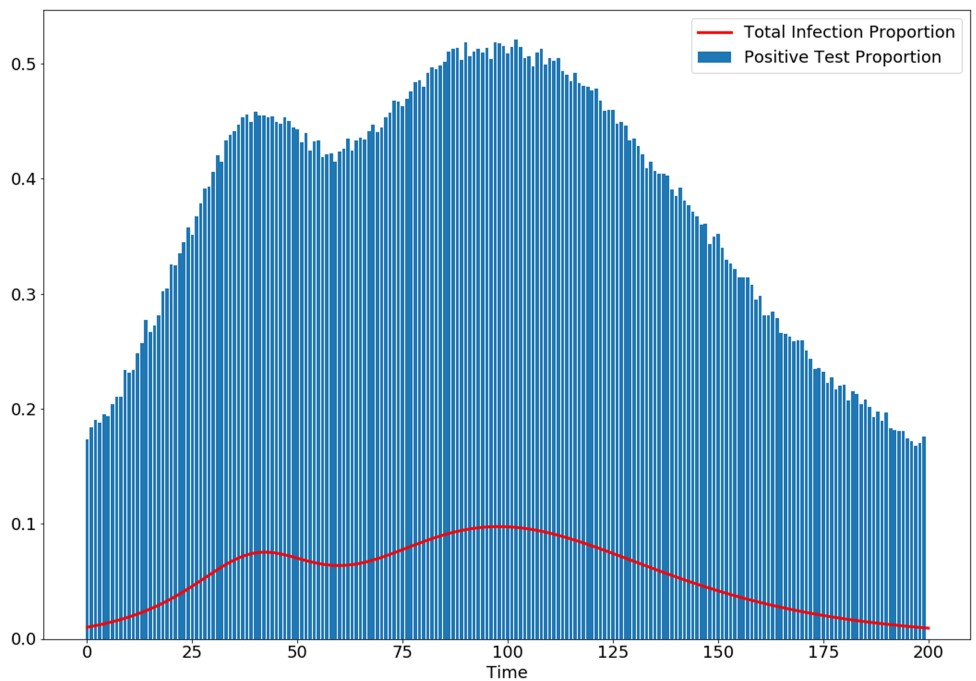

**Figure 4 Simulation shows how biasing testing towards symptomatic individuals overestimates infected proportion of the population, even with no false-positive or false-negative tests, and a high rate of testing.** Here, the bias parameter is set to $B(t) = 10$ and the testing rate is $\lambda(t) = 8,100$ for all $t$.

1,000 unbiased and 1,000 biased tests, we can estimate $B$ with a variance (and so error in the estimate of $B$) of less than 2. Depending on the magnitude of $B$, this is likely reasonable error. In simulation, this was tested with a true bias of $B = 10$, meaning that a variance of 2 represents a 20% error. In Fig. 4, we see the effect of this bias in the overestimation of the infected proportion of the population.

## Linear Trends

### Dependence on dynamic slope

The ability to correctly characterize a linear trend in the dynamics from sampled data depends on the strength of that trend as well as the nature of the population sampling. Confidence in trends is reported as the proportion of five-day intervals in 1,000 simulations which correctly identified the sign (positive/negative) of the linear fit to the dynamics. Sampling identifies the sign of the trend robustly for large enough absolute slope of the dynamics. See Appendix B for details. We also see that increasing sampling improves identification of trends in data.

In Fig. 5, we show how this dependance on the underlying dynamics effects confidence in trend identification over the course of a simulated outbreak. Here, we see that trends can be identified with good confidence with 8,100 samples per day for most time-intervals. Those intervals in which trends could not be confidently identified were those that included local maxima (peaks) or minima (valleys) in the epidemic dynamics.
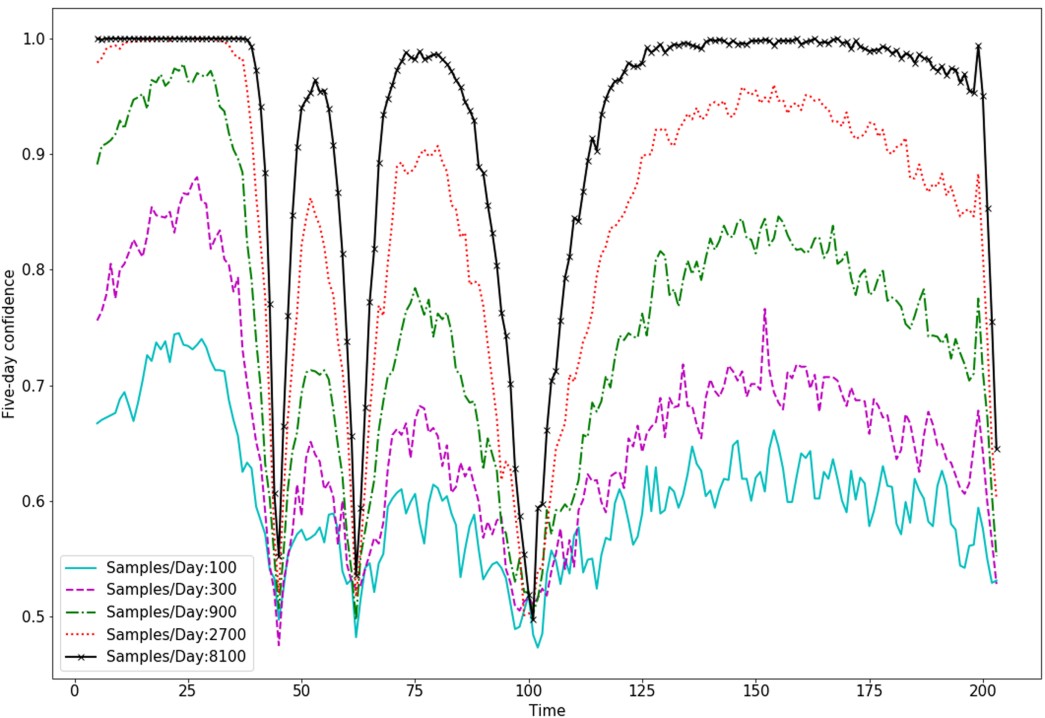

**Figure 5 Trend sign confidence changes with the underlying dynamics.**

### Confidence with bias and errors

In Fig. 6, we see that biased sampling actually improves the confidence of an estimate of the sign of a trend (i.e., is the trend positive or negative). This is because biased sampling magnifies a trend in infections which are relatively rare, meaning the trend appears stronger in the biased data. Biased testing allows for higher trend confidence in error free (no false positive/false negative) testing as well as testing with 10% error rate.

### Peak Finding

Tables 1 and 2 give precision and recall for peak finding with two sets of dynamics (SIR generated and SAIR generated) with various sampling assumptions (with and without bias and errors). We observe that identification of peaks in data using the smoothing method described above has a high chance of finding the peaks in the dynamics, but has very poor precision, providing many false peaks. See Appendix C for further details.

### Trends in COVID-19 data

Overall, our model suggests that five-day trends can be used with confidence if bias was constant for testing period. For example, we have confidence in five-day trends of the outbreak in the state of Minnesota using data from *The COVID Tracking Project* (*Lipton, Ellington & Riley, 2020*) to compute, shown in Fig. 7, with data from 6 March to 14 July 2020.

Our approach demonstrates that epidemic sampling data is more difficult to interpret accurately when the bias in testing varies with time. Unfortunately, such a variation is
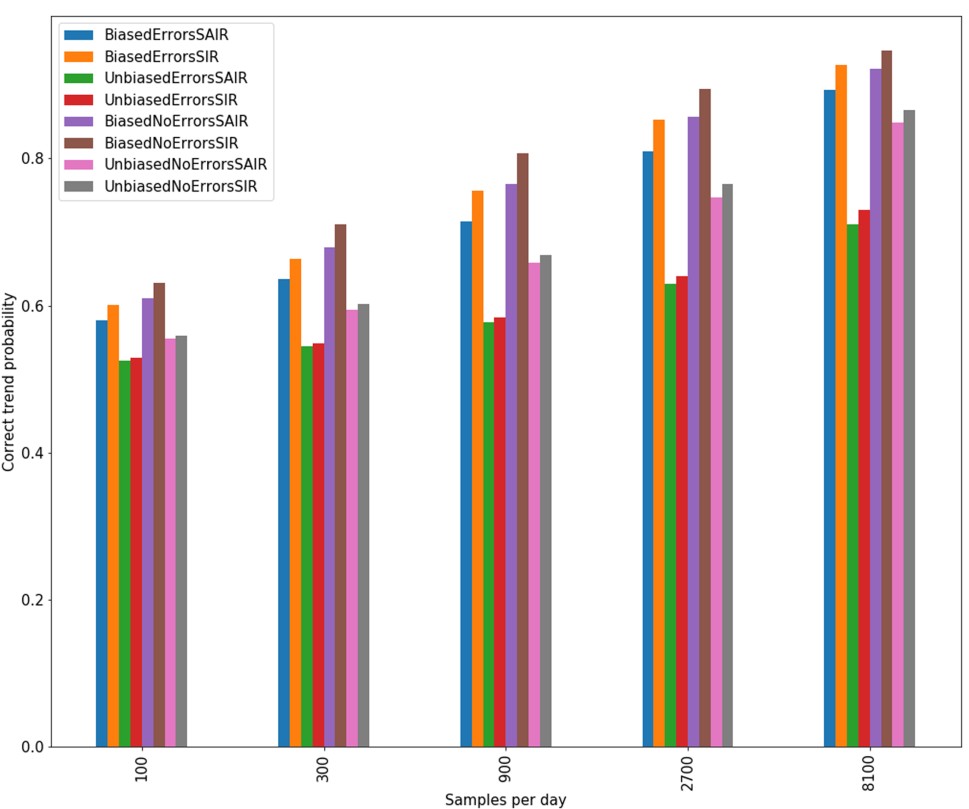

**Figure 6 Confidence in linear trends identified for sampling of SIR and SAIR dynamics with and without bias and errors.**

**Table 1 Precision of peak finding for various sampling and dynamics.**

| | Sampling per day | | |
|---|---|---|---|
| | **900** | **2,700** | **8,100** |
| BiasedSAIR | 0.1541 | 0.2198 | 0.2959 |
| BiasedSIR | 0.1865 | 0.3144 | 0.5210 |
| ExactSAIR | 0.1155 | 0.1924 | 0.3045 |
| ExactSIR | 0.1228 | 0.2080 | 0.3651 |
| PerfectSAIR | 0.1871 | 0.2566 | 0.3386 |
| PerfectSIR | 0.2400 | 0.3924 | 0.6132 |
| UnbiasedSAIR | 0.0827 | 0.1034 | 0.1568 |
| UnbiasedSIR | 0.0818 | 0.1101 | 0.1783 |

suggested by a significant negative correlation between positive test percentage and number of tests performed in many states. This can be explained by a reduction in bias as more tests become available (i.e., an increased willingness to test asymptomatic members of the population). In fact, a strong negative correlation could indicate that testing may have been initially used in a more restrictive, and therefore more heavily biased, manner. We hypothesize that as testing increases, testing bias will approach some limit that

**Table 2 Recall of peak finding for various sampling and dynamics.**

| | Sampling per day | | |
| --- | --- | --- | --- |
| | 900 | 2,700 | 8,100 |
| BiasedSAIR | 0.8955 | 0.9000 | 0.8475 |
| BiasedSIR | 0.8870 | 0.9370 | 0.9740 |
| ExactSAIR | 0.8565 | 0.9135 | 0.9370 |
| ExactSIR | 0.8580 | 0.9045 | 0.9575 |
| PerfectSAIR | 0.9060 | 0.8955 | 0.7990 |
| PerfectSIR | 0.8985 | 0.9570 | 0.9890 |
| UnbiasedSAIR | 0.8195 | 0.8505 | 0.8970 |
| UnbiasedSIR | 0.8005 | 0.8545 | 0.9180 |

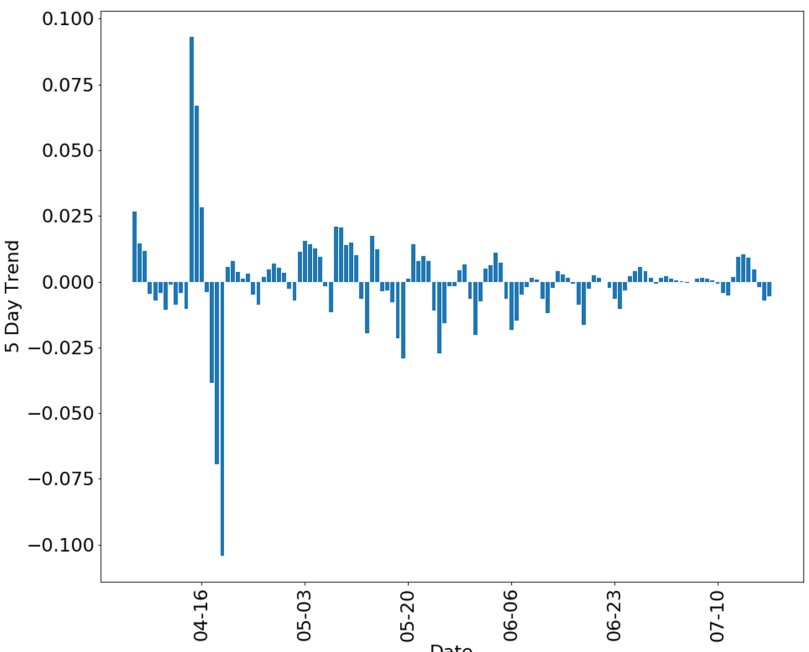

**Figure 7 Five-day trend of positive test proportion in the state of Minnesota, using data from the COVID tracking project (*Lipton, Ellington & Riley, 2020*).**

represents the preferred policies of healthcare and government organizations. It may be that even with high testing capacity, some bias will still exist due to testing practices and patient self-selection. Changes in policies will result in future changes in testing bias. We note that our model can simulate a change in bias with a time-dependent bias parameter $B$ in Eqs. (6)–(8). Our model is built with this problem in mind, allowing a time-varying total intensity function $\lambda(t)$. However, determining $\lambda(t)$ remains a challenge. This may require other than strictly testing data, such as test production data or self-reported testing bias from healthcare providers.

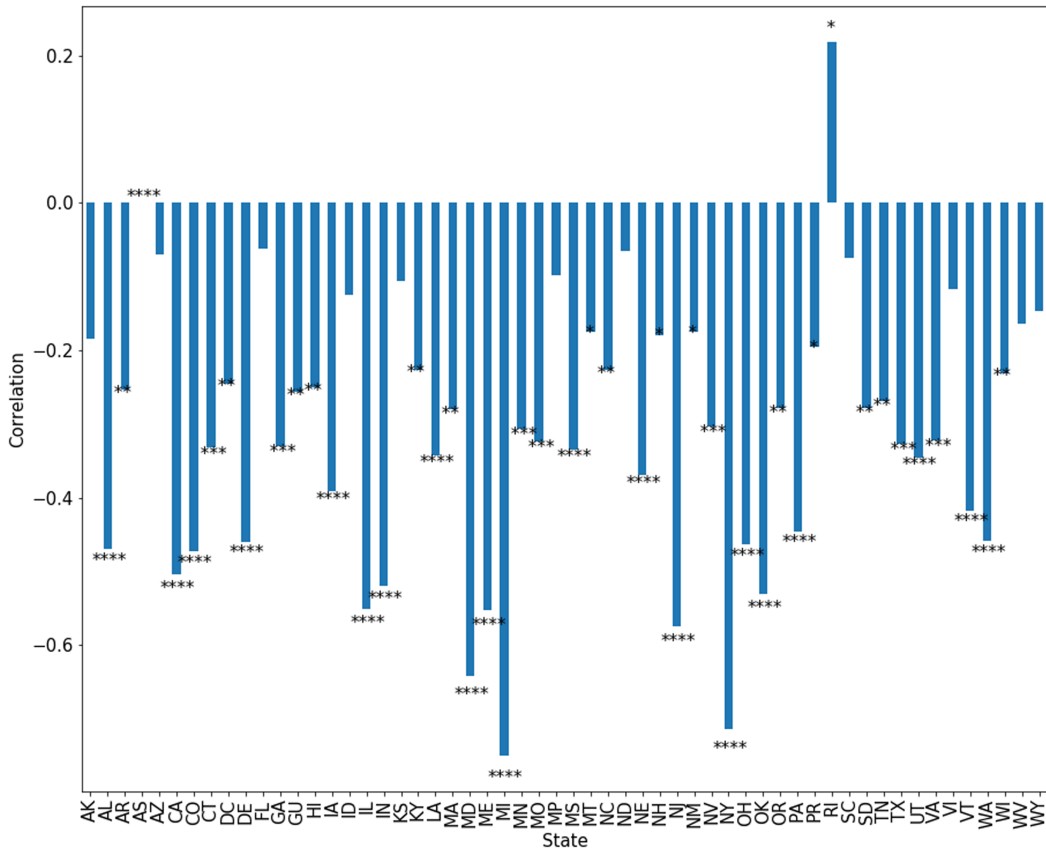

**Figure 8 Correlation between positive test percentage and tests performed in each state. Significance indicates** $^*p < 0.05$, $^{**}p < 0.01$, $^{***}p < 0.001$ **and** $^{****}p < 0.0001$.

It is worth noting that some day–day variation may be the result of irregularities in negative test reporting, as evidenced by days with 100% positive rate. Pearson correlations are shown in Fig. 8, with significance computed as $p$-value of the correlation coefficient.

## DISCUSSION

Confidence in any data analysis technique must be carefully assessed in light of the numerous confounding variables in epidemic sampling data, including bias in testing and limits to testing capacity. Our model provides realistic simulated data that can be used to assess confidence in conclusions based on sampled data, and even to calibrate and engineer novel data analysis techniques. As a relevant test of our approach and an exploration of real data from COVID-19, we use data from the *COVID Tracking Project* (*Lipton, Ellington & Riley, 2020*) for the state of Minnesota to compute five-day trends for that data, shown in Fig. 7. For data collected after mid July 2020, testing capacity was generally over 2,000 samples per day, and so these estimates can be seen as somewhat reliable with bias assumed to be approximately constant.

We also show the correlation between positive test percentage and number of tests performed for each state in Fig. 8. We see for example, that in Minnesota, this correlation is approximately −0.25. In most states, there is a significant anti-correlation between positive

test percentage and number of tests performed. This may be due to changes in the policies of healthcare providers and government organizations as tests become available. We must therefore account for this change in bias when discussing trends in the spread of the disease. Additionally, some of this correlation may be due to the occurrence of days on which negative tests are not reported or under-reported. We may model changes in bias simply by choosing some time-dependent bias functions $B(t)$ in Eqs. (6)–(8).

While the multiple purposes of infectious disease testing would be satisfied if all or almost all infected individuals could be identified, the amount of testing needing to have a high level confidence that almost all infected individuals have been identified is prohibitively high. For example, the hypergeometric distribution suggest that if we have a population of 300 million with 5% infection, then we need about 27 million unbiased tests per day for 90% confidence that we have found 90% of the cases. For COVID-19, it remains very unlikely that case numbers reported represent an accurate estimate of the extent of disease spread. Furthermore, these numbers cannot be compared from place to place or time to time because of changes in testing bias (*Lipton, Ellington & Riley, 2020*). As an example of how testing bias can affect perception of a trend, we simulate of an artificial scenario where testing capacity (i.e., $\lambda(t)$) increases drastically part-way through the course of an infection in Fig. 2, and demonstrate that considering only the number of positive tests per day completely obscures a peak in the dynamics. On the other hand, simply considering the proportion of tests in a day which are positive reveals the true dynamics. This emphasizes the importance of proportion of positive tests over the number of positive tests.

Testing for COVID-19 is clearly biased toward finding infected individuals. While reduced testing has drawbacks for addressing particular scenarios, such as screening healthcare workers, it also has important benefits for tracking the population level trends that inform policy decisions. As an intuitive example, consider a population with a very small proportion of COVID-19 cases, as would be expected in the very early or very late stages of an outbreak. Heavily biased testing helps better detect the infection by focusing on where the cases are rather than spending the vast majority of tests on negative results. In this sense, biased testing is a form of importance sampling. Furthermore, biased testing reduces the number of tests needed to identify all or most infected individuals. In Fig. 1, we show the number of samples needed for bias parameters ranging from $B = 1$ to $B = 50$. Bias in testing is the natural result of the role of testing in the healthcare setting, and this confirms the advantages bias has for detecting population trends. However, using bias in testing as shown in Eqs. (6)–(8), we see in Fig. 4 that the spread of the infection will be overestimated significantly by biased testing. In other words, to accurately estimate the spread of disease, we must estimate the bias parameter $B$. This can be done by conducting a separate set of unbiased tests and using the relationship given by Eq. (14). If we assume that the testing bias is constant (which is reasonable for a single day), this is a Monte-Carlo estimator where the error in this estimation is determined by the variance in the estimate. We simulated with a bias $B = 10$, and show the variance of single-day estimates for $B$ with various biased and unbiased testing capacities in Fig. 3. From that

simulation, we conclude that it is not unreasonable to estimate bias with 1,000 unbiased samples per day, in addition to a larger capacity of biased testing.

Policy changes during an outbreak, such as the recent activation and deactivation of stay-at-home orders in the USA, appear to be based on trends in the disease dynamics, that is, whether disease spread is accelerating or decelerating, or if there have been changes to the rate of acceleration. Our work shows that we can account for testing bias and successfully determine the underlying trend in disease dynamics. Moreover, we show that the overall positive or negative trend is not overly sensitive to the bias, meaning that assuming an approximately constant bias may work for most estimates. It is important to note that determining the sign of trends in the disease is easier when the trends are larger in magnitude, as shown in Appendix B. The less change there is in infection rate, such as those through smaller policy changes, the more testing is needed to identify an effect. As an example, we see in Fig. 5 that 8,100 samples per day is enough to give good confidence in the estimated sign of a five-day trend in disease dynamics for most of the course of an outbreak. This confidence is low when the trend is very weak, meaning the true dynamics are at a local maximum (peak) or minimum (valley). Finally, we see in Fig. 6 that a constant bias in testing actually improves our ability to detect the sign of a trend in the dynamics. This is because biased testing magnifies trends in the data, as can be seen in Fig. 4. We see again that 8,100 tests per day gives high confidence in the sign of five day trends in the data as long as that data is done with a constant high bias.

Policy may also be based on other metrics in sampling data, such as the occurrence of peaks or more complicated model fitting. Our model of sampling provides a method for testing the confidence of these metrics. As an example we show that the exact peaks in an outbreak can be found, as seen in Table 2, but there will likely be a large number of false peaks, as seen in Table 1. However, with the right smoothing, critical points in the dynamics can be identified with some confidence.

As written, our model assumes that the dynamics of an epidemic can be characterized by tracking three compartments within a society which we refer to as "symptomatic", "asymptomatic" and "healthy". Thus, any dynamics must be recast as counts of individuals who have a disease and show symptoms, those who have a disease and do not show symptoms, and those who do not have a disease. For finer grained models of epidemic sampling, these compartments can still be determined and our model used, but information may be lost. It may be beneficial then to tailor a model analogous to ours to simulate sampling on a more detailed model of epidemic sampling. This can be done simply by increasing the number of compartments in the model and calculating equations similar to Eqs. (6)–(8).

## CONCLUSION

We provide a model of sampling in a disease outbreak in order to simulate data analysis in different outbreak situations and to assess infection testing strategies. Clearly, we should account for the confidence we have in the measurements of metrics used to set policy. This confidence is affected by testing capacity, errors, and bias. Our model provides a

method to assess confidence with time-varying testing capacity and bias by simulating sampling over the course of an epidemic. This model demonstrates the importance of tracking testing capacity, estimating possible changes in bias, and tracking positive test percentage rather than raw number of positive tests. Our model provides an essential tool in designing an effective response to the outbreak of an infectious disease.

## APPENDIX A: DETAILED DESCRIPTION OF THE BIAS PARAMETER

Below, we include a calculation to more provide better intuition about the nature of the bias parameter $B$. To explain this parameter, we consider the rate at which compartments of the population are tested for a disease. In an infinitesimal time-interval $[t, t + h)$, there is some probability $p_1 h$ that an asymptomatic or healthy individual will be tested, and some probability $p_1 h$ that a symptomatic infected individual will be tested. The nature of Poisson point processes is that (assuming for simplicity no errors in the tests)

$$\lambda^+(t) = p_1 I^1(t) + p_2 I^2(t) \tag{15}$$

and

$$\lambda^-(t) = p_1 H(t) \tag{16}$$

and that the total rate of testing is $\lambda(t) = p_1(I^1(t) + H(t)) + p_2 I^2(t)$. We then have from Eq. (9) that

$$p_1 I^1(t) + p_2 I^2(t) = (p_1(I^1(t) + H(t)) + p_2 I^2(t)) \frac{I^1(t) + BI^2(t)}{I^1(t) + BI^2(t) + H(t)}. \tag{17}$$

We can rewrite this as

$$\frac{p_1 I^1(t) + p_2 I^2(t)}{(p_1(I^1(t) + H(t)) + p_2 I^2(t))} = \frac{I^1(t) + BI^2(t)}{I^1(t) + BI^2(t) + H(t)} \tag{18}$$

and see that

$$B = \frac{p_2}{p_1} \tag{19}$$

Thus the bias $B$ can be interpreted as the increased rate of testing of symptomatic individuals over asymptomatic individuals, as presumably $p_2 > p_1$. The effect of this difference is that the apparent population sampled is an adjusted version of the true population, with apparent total $I^1(t) + BI^2(t) + H(t)$ and apparent number of infected $I^1(t) + BI^2(t)$.

## APPENDIX B: SAMPLING FROM DIFFERENT UNDERLYING DYNAMICS

In Fig. A1, we test sampling's ability to identify a constant trend (i.e., linear increase or decrease) in the infected proportion. We see that sampling identifies the sign of the trend robustly for large enough absolute slope.
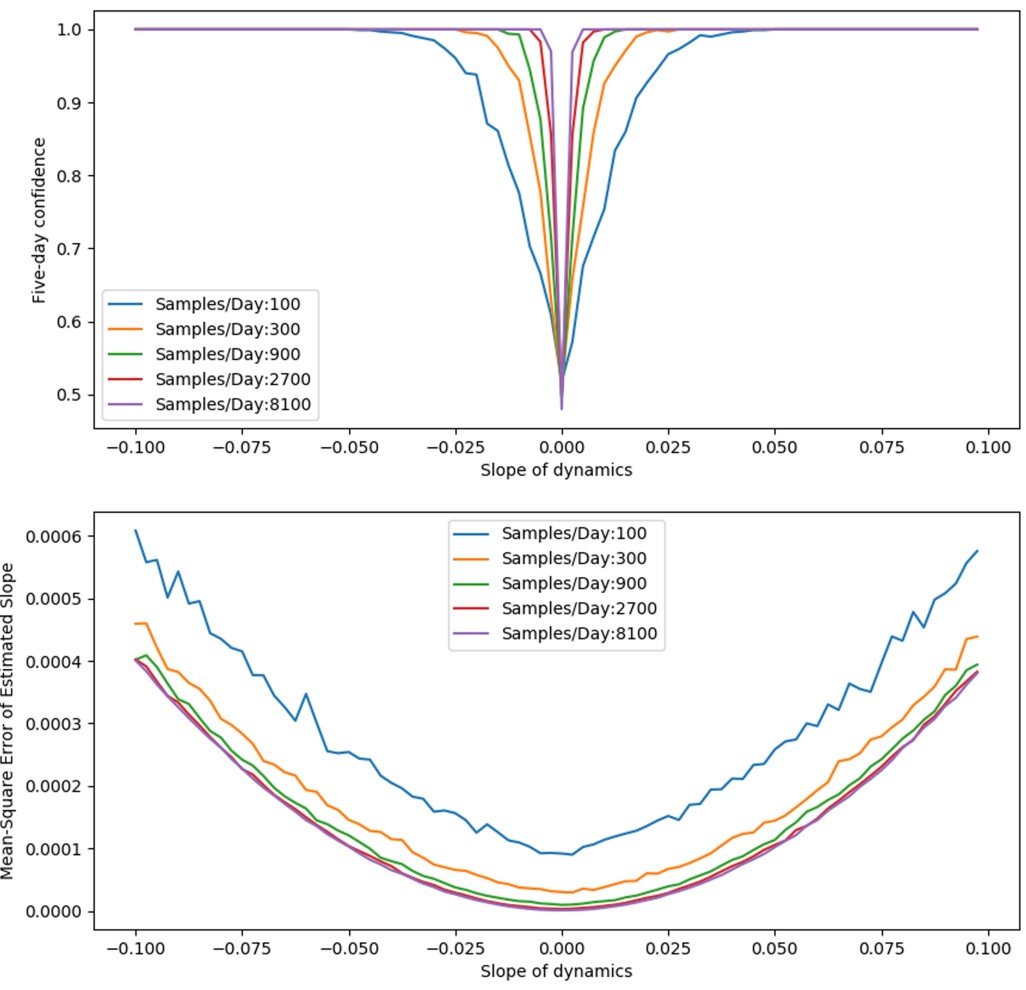

**Figure A1 Trend fitting for five-day interval of dynamics with constant slope.** Top: Confidence in the sign of the estimated slope as actual slope varies. Bottom: Error in estimated slope as slope varies.

# APPENDIX C: RESULTS OF PEAK FINDING

Tables 1 and 2 were generated with a smoothing parameter of 5. Here, we demonstrate that this can be improved with an optimal choice of smoothing. Figure A2 uses smoothing from 1 to 10, with 10 giving the highest precision and recall.

# APPENDIX D: MODELS USED TO GENERATE DYNAMICS

## SIR model

We generate SIR dynamics by considering three pools of individuals: those that are susceptible, those that are infected, and those that have recovered. Individuals transition between these pools according to mass action dynamics given in the ODE model:

$$\frac{dx_S}{dt} = -\beta x_I x_S$$

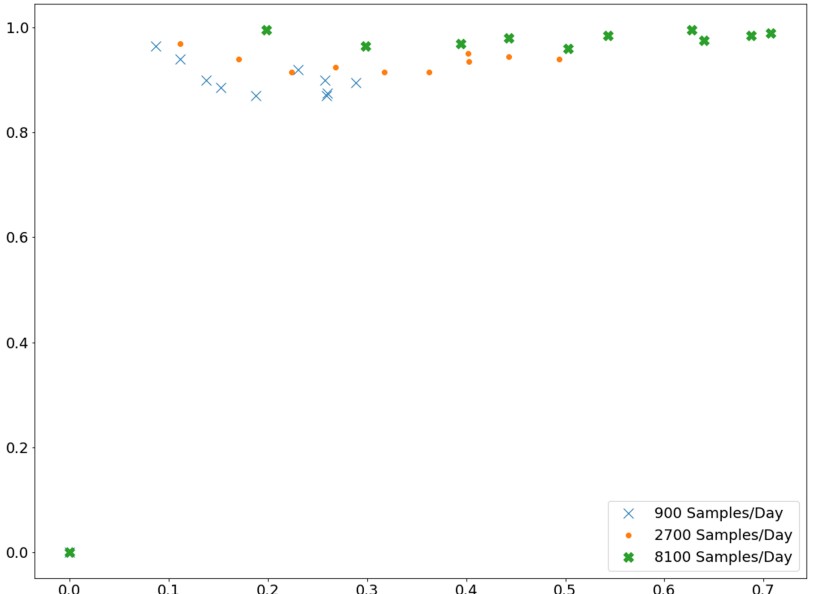

**Figure A2 Precision and recall for peak finding with various smoothing parameters.** These simulations sought peaks in a course of infection dynamics generated by the SIR model with a variable $\beta(t)$. The underlying dynamics were the same as shown in Fig. 4. Sampling was done with bias of 10 and false positive/false negative rates of 10%.

$$\frac{dx_I}{dt} = \beta x_I x_S - \gamma x_i$$

$$\frac{dx_R}{dt} = \gamma x_I$$

where $x_S$, $x_I$, $x_R$ represent the proportion of the population that is susceptible, infected, or recovered from the disease. We allow $\beta = \beta(t)$ to be a function of time, and choose a dynamic parameterization which allows us to generate an infection with more than one peak time, as can be seen in Fig. 4. We do this simply by varying $\beta$ (intuitively varying the virulence of the disease) so that it decreased until $t = 50$ and then increased, with maximum of $\frac{2}{15}$:

$$\beta(t) = \frac{1}{15}\left(2 - \exp\left(-\left(\frac{t - 50}{15}\right)^2\right)\right) \tag{20}$$

and $\gamma = \frac{1}{15}$. This dynamic parameterization allows us to generate an infection with more than one peak time, as can be seen in Fig. 4.

This model can be interpreted as stating that individuals transition from susceptible to infected at the rate $\beta x_I x_S$, and transition from infected to recovered at the rate $\gamma x_i$. The well known "$R(t)$" parameter is defined as

$$R(t) = \frac{N\beta}{\gamma} \tag{21}$$

where $N$ is the total population size (*Edelstein-Keshet, 2005*).

Using these dynamics, we take $I^1(t) = x_I(t)$, $I^2(t) = 0$ and $H(t) = x_S(t) + x_R(t)$.

## SAIR model

We also test dynamics that include an asymptomatic infected population. We generate SAIR dynamics by considering three pools of individuals: those that are susceptible, those that are infected, and those that have recovered. Individuals transition between these pools according to mass action dynamics given in the ODE model:

$$\frac{dx_S}{dt} = -(\beta_{11} + \beta_{12})x_{I^1}x_S - (\beta_{21} + \beta_{22})x_{I^2}x_S$$

$$\frac{dx_{I^1}}{dt} = \beta_{11}x_{I^1}x_S + \beta_{21}x_{I^2}x_S - \gamma x_{I^1} - \delta x_{I^1}$$

$$\frac{dx_{I^2}}{dt} = \beta_{12}x_{I^1}x_S + \beta_{22}x_{I^2}x_S - \gamma x_{I^2} + \delta x_{I^1}$$

$$\frac{dx_R}{dt} = \gamma(x_{I^1} + x_{I^2})$$

where $x_S$, $x_{I^1}$, $x_{I^2}$, $x_R$ represent the proportion of the population that is susceptible, asymptomatic infected, symptomatic infected, or recovered from the disease.

This model can be interpreted as stating that individuals transition from susceptible to asymptomatic infected at the rate $\beta_{11}x_{I^1}x_S + \beta_{21}x_{I^2}x_S$, from susceptible to symptomatic at the rate $\beta_{12}x_{I^1}x_S + \beta_{22}x_{I^2}x_S$, from asymptomatic to symptomatic at the rate $\delta x_{I^1}$, and recover at the rate $\gamma x_{I^1}$ if asymptomatic and $\gamma x_{I^2}$ if symptomatic. Note that if we take $\beta_{11} = \beta_{12} = \beta_{21} = \beta_{22}$ we may again define the intrinsic reproduction rate $R(t)$ as in the SIR model.

Using these dynamics, we have $I^1(t) = x_{I^1}(t)$, $I^2(t) = x_{I^2}(t)$ and $H(t) = x_S(t) + x_R(t)$.

# APPENDIX E: ACCOUNTING FOR SMALL POPULATIONS

The model as described above assumes that members of the population may be re-tested immediately after being tested. This is reflected in the fact that performing a test has no effect on Eqs. (9) and (10). In large populations, this assumption is reasonable because the proportion of the population who have been tested is not significant. On the other hand, in small populations we must model the limited availability of untested members of the population.

To account for this effect, we must estimate the proportion of each sub-population ($I^1(t)$, $I^2(t)$, $H(t)$) which is available for testing. We assume that testing removes one person from testable population, and those removed are re-introduced after some exponential wait time. However, we still must approximate how the overall disease dynamics change the tested and untested population. That is, we must account for a healthy individual who has been tested becoming infected before being re-introduced into the testable population, or an infected individual recovering.

We make the following simplifying assumption that the dynamics of each sub-population are distributed uniformly across the tested and untested parts of the sub-population:

$$\frac{dx_T}{dt} = \frac{x_T}{x}\frac{dx}{dt} \tag{22}$$

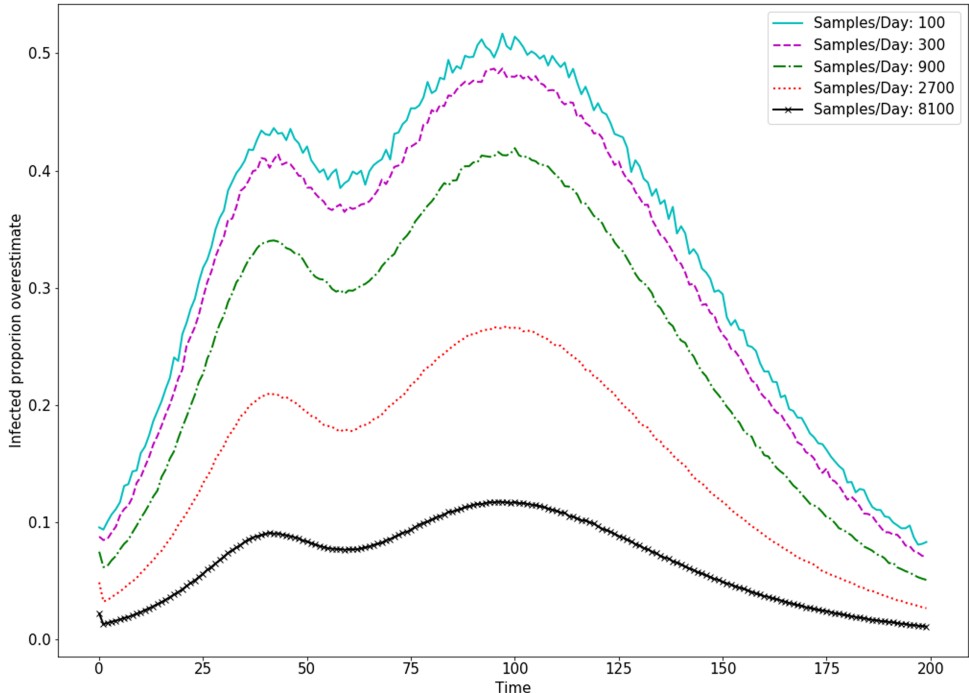

**Figure A3 Apparent bias decreases as the sampling rate increases in a limited population.
Simulations were done with a population size of 10,000.**

where $x = I^1$, $x = I^2$, or $X = H$ represent total sub-populations and $x_T$ is the number of tested and not-yet reintroduced members of the sub-population. In practice, we use an Euler approximation to estimate the proportion of a sub-population ineligible for testing:

$$x_T(t_2) \approx x_T(t_1) + \frac{x_T(t_1)}{x(t_1)}(x(t_2) - x(t_1)) = \frac{x_T(t_1)}{x(t_1)}x(t_2). \tag{23}$$

where $t_1$, $t_2$ are the times of consecutive stochastic events in the model (i.e., a test performed or population member re-introduced into the testable population).

With this model, overestimation of the positive percentage due to biased testing lessens as the rate of testing increases, as shown in Fig. A3. This is due to the limited number of testable infected individuals at any time.

### Funding
This work was supported by the Mayo Clinic Center for Individualized Medicine. There was no additional external funding received for this study. The funders had no role in study design, data collection and analysis, decision to publish, or preparation of the manuscript.

### Grant Disclosures
The following grant information was disclosed by the authors:
Mayo Clinic Center for Individualized Medicine.

## Competing Interests

The authors declare that they have no competing interests.

## Author Contributions

- James Brunner conceived and designed the experiments, performed the experiments, analyzed the data, prepared figures and/or tables, authored or reviewed drafts of the paper, and approved the final draft.
- Nicholas Chia conceived and designed the experiments, analyzed the data, authored or reviewed drafts of the paper, and approved the final draft.

## Data Availability

Our code to generate sample paths of the model is written in the language go, and compiled as the executable file disease_confidence. Code to perform Monte-Carlo simulation is available as python scripts. All code is available at GitHub: https://github.com/jdbrunner/EpidemicSampling.

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
