# Peer review of "Confidence in the dynamic spread of epidemics under biased sampling conditions"

_PeerJ, doi:10.7717/peerj.9758_

## Round 0.1 · original submission · Major Revisions

Dear authors, Please only address the comments of Reviewers 2 & 3 in your revised version. They will check it later. We will not return your revised version to Reviewer 1.

·

Basic reporting

In this manuscript, a stochastic model to simulate the sampling of a population is developed based on a class of hypergeometric distribution with population rBT+(1-r)T and infected rBT is provided, where B is the same biased used in our sampling model below, r is the proportion of the population that is infected, and T is the total population size. Such model simulates testing for a disease in an epidemic with known dynamics, allowing us to use Monte-Carlo sampling to assess metric confidence. Also, the SIR epidemic model and SAIR epidemic model are tested. The model provides an essential tool in designing an effective response to the outbreak of an infectious disease. The obtained results are interesting. The effect of precautionary measures on the behavior of the model was confirmed that the quarantine period should be long enough to achieve the desired result.

The authors should do the following to improve the quality of the manuscript:
I. The paper is well written, and it is written in a truly sporty manner. English is generally good, I think it needs to be polished further and some typos need to be revised. Further punctuation marks should be checking through the paper, especially after the equations and at the end of the statements.
II. The original papers should be much better presented.
III. References need to be arranged properly according to the journal format.
IV. The literature should be updated with the relevant bibliography "Solution of Fractional SIR Epidemic Model Using Residual Power Series Method, Applied Mathematics and Information Sciences 13 (2), 153-161 (2019)", "On the homotopy analysis method for fractional SEIR epidemic model, Research Journal of Applied Sciences, Engineering and Technology, 7 (18) 3809-3820 (2014)", "Analytical Study of Fractional-Order Multiple Chaotic FitzHugh-Nagumo Neurons Model Using Multistep Generalized Differential Transform Method, Abstract and Applied Analysis 2014, 276279 (2014)",

Experimental design

Please see the comments on basic reporting

Validity of the findings

no comment

Additional comments

no comment

Reviewer 2 ·

Basic reporting

The authors show that they have worked on the text, but additional work is needed to make it more accessible to the reader. Several sections are short and not connected, there should be a flow on the paper that leads to the main story. Additionally, I find lines 80-82 hard to understand.
There are interesting references. The authours should check the reference in line 48, I guess the author of that is World Health Organization. Also it would be good to include references in the first paragraph of the introduction, and cite the seroprevalence study performed in Spain that illustrated the fraction of asymptomatic infected by COVID-19.
There are a couple of spelling typos: L94 deponds->depends, L 144 possibly -> possible.
There are many figures. I recommend to select the most relevant and keep the rest for appendix/supplementary materials. Figure 4 is not cited between figs. 3 and 5.

Experimental design

The research question is very interesting and sound in the field.
The methods need some work. For example, I am missing a mathematical definition of \lambda in lines 91 and 92, it should be specified that equations (1,2,3) correspond to the unbiased case. Additionally, the matching between the mathematical model of epidemic spreading and the variables I_1, I_2 and H is done in the appendix, while I think that it is critical to include this in the main text. Additionally, when introducing SIR and SAIR, the compartments and the rates of transitions between them should be defined.

Validity of the findings

I am missing some discussion on the limitations of the work (for example, the absence of the exposed state in the dynamics).

Additional comments

I have additional suggestions to improve the manuscript:
Some limit cases and values of B (when it is introduced in the methods) would be really illustrative to provide the reader with references about this value.
Figure 1: if 5% of 300 million is infected and you want to find 90%, please mark with a horizontal line the limit case 0.05*0.9*300,000,000.
Lines 143-149: please specify that in this case B is constant.
Figure 3: I am missing a label in the color bar.

·

Basic reporting

no comment

Experimental design

no comment

Validity of the findings

no comment

---

## Round 0.2 · accepted · Accept

Please address also the minor text edits provided by the reviewers.

Reviewer 2 ·

Basic reporting

Thanks to the authors for addressing the reviewer's comments. From my side, there are just two minor remarks:
-Line 89: this should be rephrased to be grammatically correct: "We
therefore introduce non-dimensional a bias parameter"
-Figure 3: please change the size of the number font such that they do not overlap.

Experimental design

No comments

Validity of the findings

No comments

Additional comments

Thanks for improving the manuscript.

·

Basic reporting

no comment

Experimental design

no comment

Validity of the findings

no comment

Additional comments

no comment